# Repeated Contextual Auctions with Strategic Buyers

**Kareem Amin**
University of Pennsylvania
akareem@cis.upenn.edu

**Afshin Rostamizadeh**
Google Research
rostami@google.com

**Umar Syed**
Google Research
usyed@google.com

## Abstract

Motivated by real-time advertising exchanges, we analyze the problem of pricing inventory in a repeated posted-price auction. We consider both the cases of a *truthful* and *surplus-maximizing* buyer, where the former makes decisions myopically on every round, and the latter may strategically react to our algorithm, forgoing short-term surplus in order to trick the algorithm into setting better prices in the future. We further assume a buyer's valuation of a good is a function of a context vector that describes the good being sold. We give the first algorithm attaining sublinear ($\tilde{O}(T^{2/3})$) regret in the contextual setting against a surplus-maximizing buyer. We also extend this result to repeated second-price auctions with multiple buyers.

## 1 Introduction

A growing fraction of Internet advertising is sold through automated *real-time ad exchanges*. In a real-time ad exchange, after a visitor arrives on a webpage, information about that visitor and webpage, called the *context*, is sent to several advertisers. The advertisers then compete in an auction to win the *impression*, or the right to deliver an ad to that visitor. One of the great advantages of online advertising compared to advertising in traditional media is the presence of rich contextual information about the impression. Advertisers can be particular about whom they spend money on, and are willing to pay a premium when the right impression comes along, a process known as *targeting*. Specifically, advertisers can use context to specify which auctions they would like to participate in, as well as how much they would like to bid. These auctions are most often *second-price* auctions, wherein the winner is charged either the second highest bid or a prespecified *reserve price* (whichever is larger), and no sale occurs if the reserve price isn't cleared by one of the bids.

One side-effect of targeting, which has been studied only recently, is the tendency for such exchanges to generate many auctions that are rather uncompetitive or *thin*, in which few advertisers are willing to participate. Again, this stems from the ability of advertisers to examine information about the impression before deciding to participate. While this selectivity is clearly beneficial for advertisers, it comes at a cost to webpage publishers. Many auctions in real-time ad exchanges ultimately involve just a single bidder, in which case the publisher's revenue is entirely determined by the selection of reserve price. Although a lone advertiser may have a high valuation for the impression, a low reserve price will fail to extract this as revenue for the seller if the advertiser is the only participant in the auction.

As observed by [1], if a single buyer is repeatedly interacting with a seller, selecting revenue-maximizing reserve prices (for the seller) reduces to revenue-maximization in a repeated *posted-price* setting: On each round, the seller offers a good to the buyer at a price. The buyer observes her *value* for the good, and then either accepts or rejects the offer. The seller's price-setting algorithm is known to the buyer, and the buyer behaves to maximize her (time-discounted) cumulative *surplus*, i.e., the total difference between the buyer's value and the price on rounds where she accepts the offer. The goal of the seller is to extract nearly as much revenue from the buyer as would have been

possible if the process generating the buyer's valuations for the goods had been known to the seller before the start of the game. In [1] this goal is called minimizing *strategic regret*.

Online learning algorithms are typically designed to minimize regret in *hindsight*, which is defined as the difference between the loss of the best action and the loss of the algorithm given the observed sequence of events. Furthermore, it is assumed that the observed sequence of events are generated adversarially. However, in our setting, the buyer behaves self-interestedly, which is not necessarily the same as behaving adversarially, because the interaction between the buyer and seller is not zero-sum. A seller algorithm designed to minimize regret against an adversary can perform very suboptimally. Consider an example from [1]: a buyer who has a large valuation $v$ for every good. If the seller announces an algorithm that minimizes (standard) regret, then the buyer should respond by only accepting prices below some $\epsilon \ll v$. In hindsight, posting a price of $\epsilon$ in every round would appear to generate the most revenue for the seller given the observed sequence of buyer actions, and therefore $\epsilon T$ cumulative revenue is "no-regret". However, the seller was tricked by the strategic buyer; there was $(v - \epsilon)T$ revenue left on the table. Moreover, this is a good strategy for the buyer (it must have won the good for nearly nothing on $\Omega(T)$ rounds).

The main contribution of this paper is extending the setting described above to one where the buyer's valuations in each round are a function of some context observed by both the buyer and seller. While [1] is motivated by our same application, they imagine an overly simplistic model wherein the buyer's value is generated by drawing an independent $v_t$ from an unknown distribution $\mathcal{D}$. This ignores that $v_t$ will in reality be a function of contextual information $\mathbf{x}_t$, information that is available to the seller, and the entire reason auctions are thin to begin with (without $\mathbf{x}_t$ there would be no targeting). We give the first algorithm that attains sublinear regret in the contextual setting, against a surplus-maximizing buyer. We also note that in the non-contextual setting, regret is measured against the revenue that could have been made if $\mathcal{D}$ were known, and the single fixed optimal price were selected. Our comparator will be more challenging as we wish to compete with the best *function* (in some class) from contexts $\mathbf{x}_t$ to prices.

The rest of the paper is organized as follows. We first introduce a linear model by which values $v_t$ are derived from contexts $\mathbf{x}_t$. We then demonstrate an algorithm based on stochastic gradient descent (SGD) which achieves sublinear regret against an truthful buyer (one that accepts price $p_t$ iff $p_t \leq v_t$ on every round $t$). The analysis for the truthful buyer uses prexisting high probability bounds for SGD when minimizing strongly convex functions [15]. Our main result requires an extension of this analysis to cases in which "incorrect" gradients are occasionally observed. This lets us study a buyer that is allowed to best-respond to our algorithm, possibly rejecting offers that the truthful buyer would not, in order to receive better offers on future rounds. We also adapt our algorithm to non-linear settings via a kernelized version of the algorithm. Finally, we extend our results to second-price auctions with multiple buyers.

**Related Work:** The pricing of digital good in repeated auctions has been considered by many other authors, including [1, 10, 3, 2, 5, 11]. However, most of these papers do not consider a buyer who behaves strategically *across rounds*. Buyers either behave randomly [11], or only participate in a single round [10, 3, 2, 5], or participate in multiple rounds but only desire a single good [13, 7] and therefore, in each of these cases, are not incentivized to manipulate the seller's behavior on future rounds. In reality buyers repeatedly interact with the same seller. There is empirical evidence suggesting that buyers are not myopic, and do in fact behave strategically to induce better prices in the future [6], as well as literature studying different strategies for strategic buyers [4, 8, 9].

## 2  Preliminaries

Throughout this work, we will consider a repeated auction where at every round a single seller prices an item to sell to a single buyer (extensions to multiple buyers are discussed in Section 5). The good sold at step $t$ in the repeated auction is represented by a context (feature) vector $\mathbf{x}_t \in \mathcal{X} = \{\mathbf{x} \colon \|\mathbf{x}\|_2 \leq 1\}$ and is drawn according a fixed distribution $\mathcal{D}$, which is unknown to the seller. The good has a value $v_t$ that is a linear function of a parameter vector $\mathbf{w}^*$, also unknown to the seller, $v_t = {\mathbf{w}^*}^\top \mathbf{x}_t$ (extensions to non-linear functions of the context are considered in Section 5). We assume that $\mathbf{w}^* \in \mathcal{W} = \{\mathbf{w} \colon \|\mathbf{w}\|_2 \leq 1\}$ and also that $0 \leq {\mathbf{w}^*}^\top \mathbf{x} \leq 1$ with probability one with respect to $\mathcal{D}$.

For rounds $t = 1, \ldots, T$ the repeated posted-price auction is defined as follows: (1) The buyer and seller both observe $\mathbf{x}_t \sim \mathcal{D}$. (2) The seller offers a price $p_t$. (3) The buyer selects $a_t \in \{0, 1\}$. (4) The seller receives revenue $a_t p_t$.

Here, $a_t$ is an indicator variable that represents whether or not the buyer accepted the offered price (1 indicates yes). The goal of the seller is to select a price $p_t$ in each round $t$ such that the expected *regret* $R(T) = \mathrm{E}\left[\sum_{t=1}^{T} v_t - a_t p_t\right]$ is $o(T)$. The choice of $a_t$ will depend on the buyer's behavior. We will analyze two types of buyers in the subsequent sections of the paper: *truthful* and *surplus-maximizing* buyers, and will attempt to minimize *regret against the truthful buyer* and *regret against the surplus-maximizing buyer*. Note, the regret is the difference between the maximum revenue possible and the amount made by the algorithm that offers prices to the buyer.

## 3 Truthful Buyer

In this section we introduce the Learn-Exploit Algorithm for Pricing (LEAP), which we show has regret of the form $O(T^{2/3}\sqrt{\log(T \log(T))})$ against a *truthful* buyer. A buyer is truthful if she accepts any offered price that gives a non-negative *surplus*, which is defined as the difference between the buyer's value for the good minus the price paid: $v_t - p_t$. Therefore, for a truthful buyer we define $a_t = \mathbf{1}\{p_t \le v_t\}$.

At this point, we note that the loss function $v_t - \mathbf{1}\{p_t \le v_t\}p_t$, which we wish to minimize over all rounds, is not convex, differentiable or even continuous. If the price is even slightly above the truthful buyers valuation it is rejected and the seller makes zero revenue. To circumvent this our algorithm will attempt to learn $\mathbf{w}^*$ directly by minimizing a surrogate loss function for which $\mathbf{w}^*$ in the minimizer. Our analysis hinges on recent results [15] which give optimal rates for gradient descent when the function being minimized is strongly convex. Our key trick is to offer prices so that, in each round, the buyer's behavior reveals the gradient of the surrogate loss at our current estimate for $\mathbf{w}^*$. Below we define the LEAP algorithm (Algorithm 1), which we show addresses these difficulties in the online setting.

---

**Algorithm 1** LEAP algorithm

- Let $0 \le \alpha \le 1$, $\mathbf{w}_1 = \mathbf{0} \in \mathcal{W}$, $\epsilon \ge 0$, $\lambda > 0$, $T_\alpha = \lceil \alpha T \rceil$.
- For $t = 1, \ldots, T_\alpha$         (Learning phase)
  - Offer $p_t \sim U$, where $U$ is the uniform distribution on the interval $[0, 1]$.
  - Observe $a_t$.
  - $\tilde{\mathbf{g}}_t = 2(\mathbf{w}_t \cdot \mathbf{x}_t - a_t)\mathbf{x}_t$.
  - $\mathbf{w}_{t+1} = \Pi_{\mathcal{W}}(\mathbf{w}_t - \frac{1}{\lambda t}\tilde{\mathbf{g}}_t)$.
- For $t = T_\alpha + 1, \ldots, T$         (Exploit phase)
  - Offer $p_t = \mathbf{w}_{T_\alpha+1} \cdot \mathbf{x}_t - \epsilon$.

---

The algorithm depends on input parameters $\alpha$, $\epsilon$ and $\lambda$. The $\alpha$ parameter determines what fraction of rounds are spent in the learning phase as oppose to the exploit phase. During the learning phase, uniform random prices are offered and the model parameters are updated as a function of the feedback given by the buyer. During the exploit phase, the model parameters are fixed and the offered price is computed as a linear function of these parameters minus the value of the $\epsilon$ parameter. The $\epsilon$ parameter can be thought of as inversely proportional to our confidence in the fixed model parameters and is used to hedge against the possibility of over-estimating the value of a good. The $\lambda$ parameter is a learning-rate parameter set according to the minimum eigenvalue of the covariance matrix, and is defined below in Assumption 1. In order to prove a regret bound, we first show that the learning phase of the algorithm is minimizing a strongly convex surrogate loss and then show that this implies the buyer enjoys near optimal revenue during the exploit phase of the algorithm.

Let $\mathbf{g}_t = 2(\mathbf{w}_t^\top \mathbf{x}_t - \mathbf{1}\{p_t \le v_t\})\mathbf{x}_t$ and $F(\mathbf{w}) = \mathrm{E}_{\mathbf{x} \sim \mathcal{D}}\left[(\mathbf{w}^{*\top}\mathbf{x} - \mathbf{w}^\top \mathbf{x})^2\right]$. Note that when the buyer is truthful $\tilde{\mathbf{g}}_t = \mathbf{g}_t$. Against a truthful buyer, $\mathbf{g}_t$ is an unbiased estimate of the gradient of $F$.

**Proposition 1.** *The random variable $\mathbf{g}_t$ satisfies $\mathrm{E}[\mathbf{g}_t \mid \mathbf{w}_t] = \nabla F(\mathbf{w}_t)$. Also, $\|\mathbf{g}_t\| \le 4$ with probability 1.*

*Proof.* First note that $\mathrm{E}[\mathbf{g}_t \mid \mathbf{w}_t] = \mathrm{E}_{\mathbf{x}_t}\big[2\big(\mathbf{w}_t \cdot \mathbf{x}_t - \mathrm{E}_{p_t}[\mathbf{1}\{p_t \leq v_t\}]\big)\big] = \mathrm{E}_{\mathbf{x}_t}\big[2\big(\mathbf{w}_t \cdot \mathbf{x}_t - \mathrm{Pr}_{p_t}(p_t \leq v_t)\big)\big]$. Since $p_t$ is drawn uniformly from $[0, 1]$ and $v_t$ is guaranteed to lie in $[0, 1]$ we have that $\mathrm{Pr}(p_t \leq v_t) = \int_0^1 \mathbf{1}\{p_t \leq v_t\} dp_t = v_t$. Plugging this back into $\mathbf{g}_t$ gives us exactly the expression for $\nabla F(\mathbf{w}_t)$. Furthermore, $\|\mathbf{g}_t\| = 2|\mathbf{w}_t^\top \mathbf{x}_t - \mathbf{1}\{p_t \leq v_t\}|\,\|\mathbf{x}_t\| \leq 4$ since $|\mathbf{w}_t^\top \mathbf{x}_t| \leq \|\mathbf{w}_t\|\|\mathbf{x}_t\| \leq 1$ and $\|\mathbf{x}_t\| \leq 1$ $\qquad\square$

We now introduce the notion of *strong convexity*. A twice-differentiable function $H(\mathbf{w})$ is $\lambda$-strongly convex if and only if the Hessian matrix $\nabla^2 H(\mathbf{w})$ is full rank and the minimum eigenvalue of $\nabla^2 H(\mathbf{w})$ is at least $\lambda$. Note that the function $F$ is strongly convex if and only if the covariance matrix of the data is full-rank, since $\nabla^2 F(\mathbf{w}) = 2\mathrm{E}_{\mathbf{x}}[\mathbf{x}\mathbf{x}^\top]$. We make the following assumption.

**Assumption 1.** *The minimum eigenvalue of* $2\mathrm{E}_{\mathbf{x}}[\mathbf{x}\mathbf{x}^\top]$ *is at least* $\lambda > 0$.

Note that if this is not the case then there is redundancy in the features and the data can be projected (for example using PCA) into a lower dimensional feature space with a full-rank covariance matrix and without any loss in information. The seller can compute an offline estimate of both this projection and $\lambda$ by collecting a dataset of context vectors before starting to offer prices to the buyer.

Thus, in view of Proposition 1 and the strong convexity assumption, we see the learning phase of the LEAP algorithm is conducting a stochastic gradient descent to minimize the $\lambda$-strongly convex function $F$, where at each time step we update $\mathbf{w}_{t+1} = \Pi_{\mathcal{W}}(\mathbf{w}_t - \frac{1}{\lambda t}\tilde{\mathbf{g}}_t)$ and $\tilde{\mathbf{g}}_t = \mathbf{g}_t$ is an unbiased estimate of the gradient. We now make use of an existing bound ([14, 15]) for stochastic gradient descent on strongly convex functions.

**Lemma 1** ([14] Proposition 1). *Let* $\delta \in (0, 1/e)$, $T_\alpha \geq 4$ *and suppose* $F$ *is* $\lambda$-*strongly convex over the convex set* $\mathcal{W}$. *Also suppose* $\mathrm{E}[\mathbf{g}_t \mid \mathbf{w}_t] = \nabla F(\mathbf{w})$ *and* $\|\mathbf{g}_t\|^2 \leq G^2$ *with probability 1. Then with probability at least* $1 - \delta$ *for any* $t \leq T_\alpha$ *it holds that*

$$\|\mathbf{w}_t - \mathbf{w}^*\|^2 \leq \frac{(624 \log(\log(T_\alpha)/\delta) + 1)G^2}{\lambda^2 t} \quad \text{where} \quad \mathbf{w}^* = \mathrm{argmin}_{\mathbf{w}} F(\mathbf{w}).$$

This guarantees that, with high probability, the distance between the learned parameter vector $\mathbf{w}_t$ and the target weight vector $\mathbf{w}^*$ is bounded and decreasing as $t$ increases. This allows us to carefully tune the $\epsilon$ parameter that is used in the exploit phase of the algorithm (see Lemma 6 in the appendix). We are now equipped to prove a bound on the regret of the LEAP algorithm.

**Theorem 1.** *For any* $T > 4$, $0 < \alpha < 1$ *and assuming a truthful buyer, the LEAP algorithm with* $\epsilon = \sqrt{\frac{(624 \log(\sqrt{T_\alpha} \log(T_\alpha)) + 1)G^2}{\lambda^2 T_\alpha}}$, *where* $G = 4$, *has regret against a truthful buyer at most* $R(T) \leq 2\alpha T + 4\sqrt{\frac{T}{\alpha}} \sqrt{\frac{(624 \log(\sqrt{T_\alpha} \log(T_\alpha)) + 1)G^2}{\lambda^2}}$, *which implies for* $\alpha = T^{-1/3}$ *a regret at most*

$$R(T) \leq 2T^{2/3} + 4T^{2/3} \sqrt{\frac{(624 \log(T^{1/3} \log(T^{2/3})) + 1)G^2}{\lambda^2}} = O\left(T^{2/3} \sqrt{\log(T \log(T))}\right).$$

*Proof.* We first decompose the regret

$$\mathrm{E}\Big[\sum_{t=1}^T v_t - a_t p_t\Big] = \mathrm{E}\Big[\sum_{t=1}^{T_\alpha} v_t - a_t p_t\Big] + \mathrm{E}\Big[\sum_{t=T_\alpha+1}^T v_t - a_t p_t\Big] \leq T_\alpha + \sum_{t=T_\alpha+1}^T \mathrm{E}\Big[v_t - a_t p_t\Big], \quad (1)$$

where we have used the fact $|v_t - a_t p_t| \leq 1$. Let $A$ denote the event that, for all $t \in \{T_\alpha + 1, \ldots, T\}$, $a_t = 1 \wedge v_t - p_t \leq \epsilon$. Lemma 6 (see Appendix, Section A.1) proves that $A$ occurs with probability at least $1 - T_\alpha^{-1/2}$. For brevity let $N = \sqrt{(624 \log(\sqrt{T_\alpha} \log(T_\alpha)) + 1)G^2/\lambda^2}$, then we can decompose the expectation in the following way:

$$\mathrm{E}\Big[v_t - a_t p_t\Big] = \mathrm{Pr}[A]E[v_t - a_t p_t | A] + (1 - \mathrm{Pr}[A])E[v_t - a_t p_t | \neg A]$$

$$\leq \mathrm{Pr}[A]\epsilon + (1 - \mathrm{Pr}[A]) \leq \epsilon + T_\alpha^{-1/2} = \sqrt{\frac{N}{T_\alpha}} + \sqrt{\frac{1}{T_\alpha}} \leq 2\sqrt{\frac{N}{T_\alpha}},$$

where the inequalities follow from the definition of $A$, Lemma 6, and the fact that $|v_t - a_t p_t| < 1$. Plugging this back into equation (1) gives $T_\alpha + \sum_{t=T_\alpha+1}^T \mathrm{E}[v_t - a_t p_t] \leq T_\alpha + \frac{\lceil (1-\alpha)T \rceil}{\sqrt{T_\alpha}} 2\sqrt{N} \leq 2\alpha T + 4\sqrt{\frac{T}{\alpha}}\sqrt{N}$, proving the first result of the theorem. $\alpha = T^{-1/3}$ gives the final expression. $\qquad\square$

In the next section we consider the more challenging setting of a surplus-maximizing buyer, who may accept/reject prices in a manner meant to lower the prices offered.

## 4 Surplus-Maximizing Buyer

In the previous section we considered a truthful buyer who myopically accepts every price below her value, i.e., she sets $a_t = \mathbf{1}\{p_t \leq v_t\}$ for every round $t$. Let $S(T) = E\left[\sum_{t=1}^{T} \gamma_t a_t (v_t - p_t)\right]$ be the buyer's cumulative discounted surplus, where $\{\gamma_t\}$ is a decreasing discount sequence, with $\gamma_t \in (0,1)$. When prices are offered by the LEAP algorithm, the buyer's decisions about which prices to accept during the learning phase have an influence on the prices that she is offered in the exploit phase, and so a surplus-maximizing buyer may be able to increase her cumulative discounted surplus by occasionally behaving untruthfully. In this section we assume that the buyer knows the pricing algorithm and seeks to maximize $S(T)$.

**Assumption 2.** *The buyer is surplus-maximizing, i.e., she behaves so as to maximize $S(T)$, given the seller's pricing algorithm.*

We say that a *lie* occurs in any round $t$ where $a_t \neq \mathbf{1}\{p_t \leq v_t\}$. Note that a surplus-maximizing buyer has no reason to lie during the exploit phase, since the buyer's behavior during exploit rounds has no effect on the prices offered. Let $\mathcal{L} = \{t : 1 \leq t \leq T_\alpha \wedge a_t \neq \mathbf{1}\{p_t \leq v_t\}\}$ be the set of learning rounds where the buyer lies, and let $L = |\mathcal{L}|$ be the number of lies. Observe that $\tilde{\mathbf{g}}_t \neq \mathbf{g}_t$ in any lie round (recall that $E[\mathbf{g}_t \mid \mathbf{w}_t] = \nabla F(\mathbf{w}_t)$, i.e., $\mathbf{g}_t$ is the stochastic gradient in round $t$).

We take a moment to note the necessity of the discount factor $\gamma_t$. This essentially models the buyer as valuing surplus at the current time step more than in the future. Another way of interpreting this, is that the seller is more "patient" as compared to the buyer. In [1] the authors show a lower bound on the regret against a surplus-maximizing buyer in the contextless setting of the form $O(T_\gamma)$, where $T_\gamma = \sum_{i=1}^{T} \gamma_t$. Thus, if no decreasing discount factor is used, i.e. $\gamma_t = 1$, then sublinear regret is not possible. Note, the lower bound of the contextless setting applies here as well, since the case of a distribution $\mathcal{D}$ that induces a fixed context $\mathbf{x}^*$ on every round is a special case of our setting. In that case the problem reduces to the fixed unknown value setting since on every round $v_t = \mathbf{w}^{*\top}\mathbf{x}^*$.

In the rest of this section we prove an $O\left(T^{2/3}\sqrt{\log(T)/\log(1/\gamma)}\right)$ bound on the seller's regret under the assumption that the buyer is surplus-maximizing and that her discount sequence is $\gamma_t = \gamma^{t-1}$ for some $\gamma \in (0,1)$. The idea of the proof is to show that the buyer incurs a cost for telling lies, and therefore will not tell very many, and thus the lies she does tell will not significantly affect the seller's estimate of $\mathbf{w}^*$.

**Bounding the cost of lies:** Observe that in any learning round where the surplus-maximizing buyer tells a lie, she loses surplus in that round relative to the truthful buyer, either by accepting a price higher than her value (when $a_t = 1$ and $v_t < p_t$) or by rejecting a price less than her value (when $a_t = 0$ and $v_t > p_t$). This observation can be used to show that lies result in a substantial loss of surplus relative to the truthful buyer, provided that in most of the lie rounds there is a nontrivial gap between the buyer's value and the seller's price. Because prices are chosen uniformly at random during the learning phase, this is in fact quite likely, and with high probability the surplus lost relative to the truthful buyer during the learning phase grows exponentially with the number of lies. The precise quantity is stated in the Lemma below. A full proof appears in the appendix, Section A.3.

**Lemma 2.** *Let the discount sequence be defined as $\gamma_t = \gamma^{t-1}$ for $0 < \gamma < 1$ and assume the buyer has told $L$ lies. Then for $\delta > 0$ with probability at least $1 - \delta$ the buyer loses a surplus of at least $\frac{\gamma^{-L+3}-1}{8T_\alpha \log(\frac{1}{\delta})}\left(\frac{\gamma^{T_\alpha}}{1-\gamma}\right)$ relative to the truthful buyer during the learning phase.*

**Bounding the number of lies:** Although we argued in the previous lemma that lies during the learning phase cause the surplus-maximizing buyer to lose surplus relative to the truthful buyer, those lies may result in lower prices offered during the exploit phase, and thus the overall effect of lying may be beneficial to the buyer. However, we show that there is a limit on how large that benefit can be, and thus we have the following high-probability bound on the number of learning phase lies.

**Lemma 3.** *Let the discount sequence be defined as $\gamma_t = \gamma^{t-1}$ for $0 < \gamma < 1$. Then for $\delta > 0$ with probability at least $1 - \delta$, the number of lies $L \leq \frac{\log(32T_\alpha \frac{1}{\delta}\log(\frac{2}{\delta})+1)}{\log(1/\gamma)}$.*

The full proof is found in the appendix (Section A.4), and we provide a proof sketch here. The argument proceeds by comparing the amount of surplus lost (compared to the truthful buyer) due to telling lies in the learning phase to the amount of surplus that could hope to be gained (compared to the truthful buyer) in the exploit phase. Due to the discount factor, the surplus lost will eventually outweigh the surplus gained as the number of lies increases, implying a limit to the number of lies a surplus maximizing buyer can tell.

**Bounding the effect of lies:** In Section 3 we argued that if the buyer is truthful then, in each learning round $t$ of the LEAP algorithm, $\tilde{\mathbf{g}}_t$ is a stochastic gradient with expected value $\nabla F(\mathbf{w}_t)$. We then use the analysis of stochastic gradient descent in [14] to prove that $\mathbf{w}_{T_\alpha+1}$ converges to $\mathbf{w}^*$ (Lemma 1). However, if the buyer can lie then $\tilde{\mathbf{g}}_t$ is not necessarily the gradient and Lemma 1 no longer applies. Below we extend the analysis in Rakhlin et al. [14] to a setting where the gradient may be corrupted by lies up to $L$ times.

**Lemma 4.** *Let* $\delta \in (0, 1/e)$, $T_\alpha \geq 2$. *If the buyer tells* $L$ *lies then with probability at least* $1 - \delta$,
$\|\mathbf{w}_{T_\alpha+1} - \mathbf{w}^*\|^2 \leq \frac{1}{T_\alpha+1}\left(\frac{(624\log(\log(T_\alpha)/\delta)+e^2)G^2}{\lambda^2} + \frac{4e^2 L}{\lambda}\right)$.

The proof of the lemma is similar to that of Lemma 1, but with extra steps needed to bound the additional error introduced due to the erroneous gradients. Due to space constraints, we present the proof in the appendix, Section A.6. Note that, modulo constants, the bound only differs by the additive term $L/T_\alpha$. That is, there is an extra additive error term that depends on the ratio of lies to number of learning rounds. Thus, if no lies are told, then there is no additive error. While if many lies are told, e.g. $L = T_\alpha$, then the bound become vacuous.

**Main result:** We are now ready to prove an upper bound on the regret of the LEAP algorithm when the buyer is surplus-maximizing.

**Theorem 2.** *For any* $0 < \alpha < 1$ *(such that* $T_\alpha \geq 4$*),* $0 < \gamma < 1$ *and assuming a surplus-maximizing buyer with exponential discounting factor* $\gamma_t = \gamma^{t-1}$*, then the LEAP algorithm using parameter* $\epsilon = \sqrt{\frac{1}{T_\alpha}\left(\frac{(624\log(2\sqrt{T_\alpha}\log(T_\alpha))+e^2)G^2}{\lambda^2} + \frac{4e^2\log(128\sqrt{T_\alpha}\log(4\sqrt{T_\alpha})+1)}{\lambda\log(1/\gamma)}\right)}$*, where* $G = 4$*, has regret against a surplus-maximizing buyer at most*

$$R(T) \leq 2\alpha T + 4\sqrt{\frac{T}{\alpha}}\sqrt{\frac{(624\log(2\sqrt{T_\alpha}\log(T_\alpha))+e^2)G^2}{\lambda^2} + \frac{4e^2\log(128\sqrt{T_\alpha}\log(4\sqrt{T_\alpha})+1)}{\lambda\log(1/\gamma)}},$$

*which for* $\alpha = T^{-1/3}$ *implies* $R(T) \leq O\left(T^{2/3}\sqrt{\frac{\log(T)}{\log(1/\gamma)}}\right)$.

*Proof.* Taking the high probability statements of Lemma 3 and Lemma 4 with $\delta/2 \in [0, 1/e]$ tells us that with probability at least $1 - \delta$, $\|\mathbf{w}_{T_\alpha} - \mathbf{w}^*\|^2 \leq \frac{1}{T_\alpha}\left(\frac{(624\log(2\log(T_\alpha)/\delta)+e^2)G^2}{\lambda^2} + \frac{4e^2\log(64T_\alpha\frac{1}{\delta}\log(\frac{4}{\delta})+1)}{\lambda\log(1/\gamma)}\right)$.

Since we assume $T_\alpha \geq 4$, if we set $\delta = T_\alpha^{-1/2}$ it implies $\delta/2 = T_\alpha^{-1/2}/2 \leq 1/e$, which is required for Lemma 4 to hold. Thus, if we set the algorithm parameter $\epsilon$ as indicated in the statement of theorem, we have that with probability at least $1 - T_\alpha^{-1/2}$ for all $t \in \{T_\alpha + 1, \ldots, T\}$ that $a_t = 1$ and $v_t - p_t \leq \epsilon$, which follows from the same argument used for Lemma 6.

Finally, the same steps as in the proof of Theorem 1 we can be used to show the first inequality. Setting $\alpha = T^{-1/3}$ shows the second inequality and completes the theorem. $\square$

Note that the bound shows that if $\gamma \to 1$ (i.e. no discounting) the bound becomes vacuous, which is to be expected since the $\Omega(T_\gamma)$ lower bound on regret demonstrates the necessity of a discounting factor. If $\gamma \to 0$ (i.e. buyer become myopic, thereby truthful), then we retrieve the truthful bound modulo constants. Thus for any $\gamma < 1$, we have shown the first sublinear bound on the seller's regret against a surplus-maximizing buyer in the contextual setting.

# 5 Extensions

**Doubling trick:** A drawback of Theorem 2 is that optimally tuning the parameters $\epsilon$ and $\alpha$ requires knowledge of the horizon $T$. The usual way of handling this problem in the standard online learning setting is to apply the 'doubling trick': If a learning algorithm that requires knowledge of $T$ has regret $O(T^c)$ for some constant $c$, then running independent instances of the algorithm during consecutive phases of exponentially increasing length (i.e., the $i$th phase has length $2^i$) will also have regret $O(T^c)$. We can also apply the doubling trick to our strategic setting, but we must exercise caution and argue that running the algorithm in phases does not affect the behavior of a surplus-maximizing buyer in a way that invalidates the proof of Theorem 2. We formally state and prove the relevant corollary in Section A.8 of the Appendix.

**Kernelized Algorithm:** In some cases, assuming that the value of a buyer is a linear function of the context may not be accurate. In this section we briefly introduce a kernelized version of LEAP, which allows for a non-linear model of the buyer value as a function of the context $x$. At the same time, the regret guarantees provided in the previous sections still apply since we can view the model as linear function of the induced features $\phi(x)$, where $\phi(\cdot)$ is a non-linear map and the kernel function $K$ is used to compute the inner product in this induced feature space: $K(x, x') = \phi(x)^\top \phi(x')$. For a more complete discussion of kernel methods see, for example, [12, 16]. For what follows, we define the projection operation $\Pi_K\big(\boldsymbol{\beta}, (\mathbf{x}_1, \ldots, \mathbf{x}_t)\big) = \boldsymbol{\beta}/\sqrt{\sum_{i,j=1}^t \beta_i \beta_j K(\mathbf{x}_i, \mathbf{x}_j)}$. The proof of Proposition 2 is moved to the appendix (Section A.7) in interest of space.

---

**Algorithm 2** Kernelized LEAP algorithm

- Let $K(\cdot, \cdot)$ be a PDS function s.t. $\forall \mathbf{x} : |K(\mathbf{x}, \mathbf{x})| \le 1$, $0 \le \alpha \le 1$, $T_\alpha = \lceil \alpha T \rceil$, $\boldsymbol{\beta} = \mathbf{0} \in \mathbb{R}^{T_\alpha}$, $\epsilon \ge 0, \lambda > 0$.
- For $t = 1, \ldots, T_\alpha$
  - Offer $p_t \sim U$
  - Observe $a_t$
  - $\beta_t = -\frac{2}{\lambda t}\big(\sum_{i=1}^{t-1} \beta_i K(\mathbf{x}_i, \mathbf{x}_t) - a_t\big)$
  - $\boldsymbol{\beta} = \Pi_K\big(\boldsymbol{\beta}, (\mathbf{x}_1, \ldots, \mathbf{x}_t)\big)$
- For $t = T_\alpha + 1, \ldots, T$
  - Offer $p_t = \sum_{i=1}^{T_\alpha} \beta_i K(\mathbf{x}_i, \mathbf{x}_t) - \epsilon$

---

**Proposition 2.** *Algorithm 2 is a kernelized implementation of the LEAP algorithm with $\mathcal{W} = \{\mathbf{w} : \|\mathbf{w}\|_2 \le 1\}$ and $\mathbf{w}_1 = \mathbf{0}$. Furthermore, if we consider the feature space induced by the kernel $K$ via an explicit mapping $\phi(\cdot)$, the learned linear hypothesis is represented as $\mathbf{w}_t = \sum_{i=1}^{t-1} \beta_i \phi(\mathbf{x}_i)$ which satisfies $\|\mathbf{w}_t\| = \sum_{i,j=1}^{t-1} \beta_i \beta_j K(\mathbf{x}_i, \mathbf{x}_j) \le 1$. The gradient is $\mathbf{g}_t = 2\big(\sum_{i=1}^{t-1} \beta_i \phi(\mathbf{x}_i)^\top \phi(\mathbf{x}_t) - a_t\big)\phi(\mathbf{x}_t)$, and $\|\mathbf{g}_t\| \le 4$.*

**Multiple Buyers:** So far we have assumed that the seller is interacting with a single buyer across multiple posted price auctions. Recall that the motivation for considering this setting was repeated *second price* auctions against a *single* buyer, a situation that happens often in online advertising because of targetting. One might nevertheless wonder whether the algorithm can be applied to a setting where there can be multiple buyers, and whether it remains robust in such a setting. We describe a way in which the analysis for the posted-price setting can carry over to multiple buyers. .

Formally, suppose there are $K$ buyers, and on round $t$, buyer $k$ receives a valuation of $v_{k,t}$. We let $k^{\mathrm{val}}(t) = \arg\max_k v_{k,t}$, $v_t^+ = v_{k^{\mathrm{val}}(t),t}$, and $v_t^- = \max_{k \ne k^{\mathrm{val}}(t)} v_{k,t}$: the buyer with the highest valuation, the highest valuation itself, and the second-highest valuation respectively. In a second price auction, each buyer also submits a bid $b_{k,t}$, and we define $k^{\mathrm{bid}}(t)$, $b_t^+$ and $b_t^-$ analogously to $k^{\mathrm{val}}(t)$, $v_t^+, v_t^-$, corresponding to the highest bidder, the largest bid, and the second-largest bid. After the seller announces a reserve price $p_t$, buyers submit their bids $\{b_{k,t}\}$, and the seller receives round $t$ revenue of $r_t = \mathbf{1}\{p_t \le b_t^+\} \max\{b_t^-, p_t\}$. The goal of the seller is to minimize $R(T) = \mathrm{E}[\sum_{t=1}^T v_t^+ - r_t]$. We assume that buyers are surplus-maximizing, and select a strategy that maps previous reserve prices $p_1, ..., p_{t-1}, p_t$, and $v_{k,t}$ to a choice of bid on round $t$.

We call $v_t^+$ the *market valuation* for good $t$. The key to extending the LEAP algorithm to the multiple buyer setting will be to treat market valuations in the same way we treated the individual buyer's valuation in the single-buyer setting. In order to do so, we make an analogous modelling assumption to that of Section 2. Specifically, we assume that there is some $\mathbf{w}^*$ such that $v_t^+ = \mathbf{w}^*{}_t^\top \mathbf{x}_t$.[1] Note that we assume a model on the *market price* itself.

At first glance, this might seem like a strange assumption since $v_t^+$ is itself the result of a maximization over $v_{k,t}$. However, we argue that it's actually rather unrestrictive. In fact the individual valuations $v_{k,t}$ can be generated arbitrarily so long as $v_{k,t} \leq \mathbf{w}^*{}_t^\top \mathbf{x}_t$ and equality holds for some $k$. In other words, we can imagine that nature first computes the market valuation $v_t^+$, then arbitrarily (even adversarialy) selects which buyer gets this valuation, and the other buyer valuations.

Now we can define $a_t = \mathbf{1}\{p_t \leq b_t^+\}$, whether the largest bid was greater than the reserve, and consider running the LEAP algorithm, but with this choice of $a_t$. Notice that for any $t$, $a_t p_t \leq r_t$, thereby giving us the following key fact: $R(T) \leq R'(T) \triangleq \mathrm{E}[\sum_{t=1}^{T} v_t^+ - a_t p_t]$. We also redefine $L$ to be the number of *market lies*: rounds $t \leq T_\alpha$ where $a_t \neq \mathbf{1}\{p_t \leq v_t^+\}$. Note the market tells a lie if either all valuations were below $p_t$, but somebody bid over $p_t$ anyway, or if some valuation was above $p_t$ but no buyer decided to outbid $p_t$. With this choice of $L$, Lemma 4 holds exactly as written but in the multiple buyer setting.

It's well-known [17] that single-shot second price auctions are *strategy-proof*. Therefore, during the exploit phase of the algorithm, all buyers are incentivized to bid truthfully. Thus, in order to bound $R'(T)$ and therefore $R(T)$, we need only rederive Lemma 3 to bound the number of *market lies*. We begin partitioning the market lies. Let $\mathcal{L} = \{t : t \leq T_\alpha, \mathbf{1}\{p_t \leq v_t^+\} \neq \mathbf{1}\{p_t \leq b_t^+\}\}$, while letting $\mathcal{L}_k = \{t : t \leq T_\alpha, v_t^+ < p_t^+ \leq b_t^+, k^{\mathrm{bid}}(t) = k\} \cup \{t \leq T_\alpha, b_t^+ < p_t \leq v_t^+, k^{\mathrm{val}}(t) = k\}$. In other words, we attribute a lie to buyer $k$ if (1) the reserve was larger than the market value, but buyer $k$ won the auction anyway, or (2) buyer $k$ had the largest valuation, but nobody cleared the reserve. Checking that $\mathcal{L} = \cup_k \mathcal{L}_k$ and letting $L_k = |\mathcal{L}_k|$ tells us that $L \leq \sum_{k=1}^{K} L_k$. Furthermore, we can bound $L_k$ using nearly identical arguments to the posted price setting, giving us the subsequent Corollary for the multiple buyer setting.

**Lemma 5.** *Let the discount sequence be defined as $\gamma_t = \gamma^{t-1}$ for $0 < \gamma < 1$. Then for $\delta > 0$ with probability at least $1 - \delta$, $L_k \leq \frac{\log(32 T_\alpha / \delta + 1)}{\log(1/\gamma)}$, and $L \leq K L_k$.*

*Proof.* We first consider the surplus buyer $k$ loses during learning rounds, compared to if he had been truthful. Suppose buyer $k$ unilaterally switches to always bidding his value (i.e. $b_{k,t} = v_{k,t}$). For a single-shot second price auction, being truthful is a dominant strategy and so he would only increase his surplus on learning rounds. Furthermore, on each round in $\mathcal{L}_k$ he would increase his (undiscounted) surplus by at least $|v_{k,t} - p_t|$. Now the analysis follows as in Lemmas 2 and 3. □

**Corollary 1.** *In the multiple surplus-maximizing buyers setting the LEAP algorithm with $\alpha = T^{-1/3}$, $\epsilon = \sqrt{\frac{1}{T_\alpha}\left(\frac{(624 \log(2\sqrt{T_\alpha}\log(T_\alpha)) + e^2)G^2}{\lambda^2} + \frac{4e^2 K \log(128\sqrt{T_\alpha}\log(4\sqrt{T_\alpha}) + 1)}{\lambda \log(1/\gamma)}\right)}$, has regret $R(T) \leq R'(T) \leq O\left(T^{2/3}\sqrt{\frac{K \log(T)}{\log(1/\gamma)}}\right)$*

## 6    Conclusion

In this work, we have introduced the scenario of contextual auctions in the presence of surplus-maximizing buyers and have presented an algorithm that is able to achieve sublinear regret in this setting, assuming buyers receive a discounted surplus. Once again, we stress the importance of the contextual setting, as it contributes to the rise of targeted bids that result in auction with single high-bidders, essentially reducing the auction to the posted-price scenario studied in this paper. Future directions for extending this work include considering different surplus discount rates as well as understanding whether small modifications to standard contextual online learning algorithms can lead to no-strategic-regret guarantees.

## Footnotes

[1]Note that we could also apply the kernelized LEAP algorithm (Algorithm 2) in the multiple buyer setting.

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
