[Supplementary Material]

# A  Appendix

## A.1  Selecting the $\epsilon$ parameter

**Lemma 6.** *Assume $T_\alpha \geq 4$. Then using the LEAP algorithm, in the presence of a truthful buyer, ensures that with probability at least $1 - T_\alpha^{-1/2}$ for all $t \in \{T_\alpha + 1, \ldots, T\}$ we have $a_t = 1$ and $v_t - p_t \leq \epsilon = \sqrt{\frac{(624 \log(\sqrt{T_\alpha} \log(T_\alpha)) + 1)G^2}{\lambda^2 T_\alpha}}$.*

*Proof.* Using Lemma 1, we have with probability at least $1 - T_\alpha^{-1/2}$ for $\mathbf{x} \in \mathcal{X}$

$$|\mathbf{w}^* \cdot \mathbf{x} - \mathbf{w}_{T_\alpha} \cdot \mathbf{x}| = |(\mathbf{w}^* - \mathbf{w}_{T_\alpha}) \cdot \mathbf{x}| \leq \|\mathbf{w}^* - \mathbf{w}_{T_\alpha}\| \|\mathbf{x}\| \leq \|\mathbf{w}^* - \mathbf{w}_{T_\alpha}\| \leq \sqrt{\frac{(624 \log(\sqrt{T_\alpha} \log(T_\alpha)) + 1)G^2}{\lambda^2 T_\alpha}} .$$

Therefore with probability $1 - T_\alpha^{-1/2}$ for all $t \in \{T_\alpha + 1, \ldots, T\}$

$$\mathbf{w}^* \cdot \mathbf{x}_t - \mathbf{w}_{T_\alpha} \cdot \mathbf{x}_t + \epsilon \geq 0 \iff a_t = 1 \quad \text{and} \quad \mathbf{w}^* \cdot \mathbf{x}_t - \mathbf{w}_{T_\alpha} \cdot \mathbf{x}_t - \epsilon \leq 0 \iff v_t - p_t \leq \epsilon,$$

which completes the lemma. $\qquad\square$

## A.2  Chernoff-style bound.

**Lemma 7.** *Let $S = \sum_{i=1}^{n} x_i$, where each $x_i \in \{0, 1\}$ is an independent random variable. Then the following inequality holds for any $0 < \epsilon < 1$.*

$$\Pr(S > (1 + \epsilon)\mathrm{E}[S]) \leq \frac{e^{\epsilon \mathrm{E}[S]}}{(1 + \epsilon)^{(1+\epsilon)\mathrm{E}[S]}} \leq \exp\left(\frac{-\epsilon^2 \mathrm{E}[S]}{4}\right).$$

*Proof.* In what follows denote $\Pr(x_i = 1) = p_i$. To show the first inequality, we follow standard steps for arriving at a multiplicative Chernoff bound. For any $t > 0$ and using Markov's inequality, we have

$$\Pr(S > (1 + \epsilon)\mathrm{E}[S]) = \Pr(\exp(tS) > \exp(t(1 + \epsilon)\mathrm{E}[S])) \leq \frac{\mathrm{E}[\exp(tS)]}{\exp(t(1 + \epsilon)\mathrm{E}[S])}. \tag{2}$$

Now, noting that the random variables are independent, the numerator of this expression can be bounded as follows

$$\mathrm{E}[\exp(tS)] = \mathrm{E}\left[\prod_{i=1}^{n} \exp(tx_i)\right] = \prod_{i=1}^{n} \mathrm{E}[\exp(tx_i)] = \prod_{i=1}^{n} p_i e^t + (1 - p_i) = \prod_{i=1}^{n} p_i(e^t - 1) + 1$$

$$\leq \prod_{i=1}^{n} \exp(p_i(e^t - 1)) = \exp\left((e^t - 1) \sum_{i=1}^{n} p_i\right) = \exp((e^t - 1)\mathrm{E}[S]),$$

where the inequality uses the fact $1 + x \leq e^x$. Plugging this back into (2) and setting $t = \log(1 + \epsilon)$ results in

$$\Pr(S > (1 + \epsilon)\mathrm{E}[S]) \leq \frac{\exp((e^t - 1)\mathrm{E}[S])}{\exp(t(1 + \epsilon)\mathrm{E}[S])} = \frac{\exp((1 + \epsilon - 1)\mathrm{E}[S])}{(1 + \epsilon)^{(1+\epsilon)\mathrm{E}[S]}} = \frac{e^{\epsilon \mathrm{E}[S]}}{(1 + \epsilon)^{(1+\epsilon)\mathrm{E}[S]}},$$

which proves the first inequality. To prove the second inequality, it suffices to show that

$$(1 + \epsilon)^{-(1+\epsilon)\mathrm{E}[S]} = \exp(-\log(1 + \epsilon)(1 + \epsilon)\mathrm{E}[S]) \leq \exp\left(-\epsilon \mathrm{E}[S] - \frac{\epsilon^2 \mathrm{E}[S]}{4}\right)$$

$$\iff \log(1 + \epsilon)(1 + \epsilon) \geq \epsilon + \frac{\epsilon^2}{4}. \tag{3}$$

To prove this, note that for $f(\epsilon) = \log(1 + \epsilon)(1 + \epsilon) - \epsilon - \epsilon^2/4$, we have

$$f(0) = 0$$

$$\forall \epsilon \in [0, 1], \ f'(\epsilon) = \log(1 + \epsilon) - \epsilon/2 \geq \epsilon - \epsilon^2/2 - \epsilon/2 > 0.$$

Thus, the function $f$ is zero at zero and increasing between values zero and one, implying it is positive between values zero and one and which proves the inequality in (3) and completes the lemma. $\qquad\square$

## A.3 Proof of Lemma 2

Before we present the proof of Lemma 2 we define a couple variables and also present an intermediate lemma. Define the variable

$$M_\rho = \sum_{t=1}^{T_\alpha} \mathbf{1}\{|v_t - p_t| < \rho\}, \tag{4}$$

as the number of times that the gap between the price offered and the buyer's value is less than $\rho$. For $\delta > 0$, let

$$\mathcal{E}_{\delta,\rho} = \left\{ M_\rho \leq 2\rho T_\alpha + \sqrt{8\rho T_\alpha \log \frac{1}{\delta}} \right\}, \tag{5}$$

denote the event that there are not too many rounds on which this gap is smaller than $\rho$. We first prove the following lemma:

**Lemma 8.** *For any $\delta > 0$ and $0 < \rho < 1$ we have $P(\mathcal{E}_{\delta,\rho}) \geq 1 - \delta$.*

*Proof.* First notice that on lie rounds, the (undiscounted) surplus lost compared to the truthful buyer is

$$\underbrace{\mathbf{1}\{p_t \leq v_t\}(v_t - p_t)}_{\text{truthful surplus}} - \underbrace{\mathbf{1}\{p_t > v_t\}(v_t - p_t)}_{\text{untruthful surplus}} = |v_t - p_t|.$$

Since each value $v_t \in [0,1]$ and price $p_t \in [0,1]$ is chosen i.i.d. during the first $T_\alpha$ rounds of the algorithm and furthermore $p_t$ is chosen uniformly at random, we have that on any round $\Pr(|v_t - p_t| < \rho) \leq 2\rho$. Using this, we note

$$\mathrm{E}[M_\rho] = \mathrm{E}\left[\sum_{t=1}^{T_\alpha} \mathbf{1}\{|v_t - p_t| < \rho\}\right] = \sum_{t=1}^{T_\alpha} \mathrm{E}[\mathbf{1}\{|v_t - p_t| < \rho\}] = \sum_{t=1}^{T_\alpha} \Pr(|v_t - p_t| < \rho) \leq 2\rho T_\alpha.$$

Now, since $M_\rho$ is a sum of $T_\alpha$ independent random variables taking values in $\{0,1\}$, Lemma 7 (in the appendix) implies

$$\Pr[M_\rho \geq (1+\epsilon)\mathrm{E}[M_\rho]] \leq \exp\left(\frac{-\epsilon^2 \mathrm{E}[M_\rho]}{4}\right).$$

After setting the right hand side equal to $\delta$ and solving for $\epsilon$, we have with probability at least $1 - \delta$,

$$M_\rho \leq \mathrm{E}[M_\rho]\left(1 + \sqrt{\frac{4}{\mathrm{E}[M_\rho]} \log \frac{1}{\delta}}\right) = \mathrm{E}[M_\rho] + \sqrt{4\mathrm{E}[M_\rho] \log \frac{1}{\delta}} \leq 2\rho T_\alpha + \sqrt{8\rho T_\alpha \log \frac{1}{\delta}},$$

which completes the proof of the intermediate lemma. $\qquad\square$

We can now give the proof of Lemma 2, which shows if we select

$$\rho^* = 1/(8T_\alpha \log(1/\delta)), \tag{6}$$

and the event $\mathcal{E}_{\delta,\rho^*}$ occurs, then at least $\frac{\gamma^{-L+3}-1}{8T_\alpha \log(\frac{1}{\delta})}\left(\frac{\gamma^{T_\alpha}}{1-\gamma}\right)$ surplus is lost compared to the truthful buyer.

*Proof of Lemma 2.* Let $M' = \left\lceil 2\rho T_\alpha + \sqrt{8\rho T_\alpha \log 1/\delta} \right\rceil$. Lemma 8 guarantees that with at least probability $1 - \delta$, $M'$ is the maximum number of rounds where $|v_t - p_t| \leq \rho$ occurs. Thus, on at least $L_\rho = L - M'$ of the lie rounds, at least $\rho$ (undiscounted) surplus is lost compared to the truthful buyer. Let $\mathcal{L}_\rho$ denote the set of rounds where these events occur (so that $|\mathcal{L}_\rho| = L_\rho$), then since the discount sequence is decreasing the disounted surplus lost is at least

$$\sum_{t \in \mathcal{L}_\rho} \gamma_t |v_t - p_t| \geq \rho \sum_{t \in \mathcal{L}_\rho} \gamma_t \geq \rho \sum_{t=T_\alpha - L_\rho}^{T_\alpha} \gamma_t.$$

We can continue to lower bound this quantity:

$$\sum_{t=T_\alpha-L_\rho}^{T_\alpha} \gamma_t \geq \sum_{t=0}^{T_\alpha-1} \gamma^t - \sum_{t=0}^{T_\alpha-L_\rho-1} \gamma^t = \frac{1-\gamma^{T_\alpha}}{1-\gamma} - \frac{1-\gamma^{T_\alpha-L_\rho}}{1-\gamma} = (\gamma^{-L_\rho}-1)\frac{\gamma^{T_\alpha}}{1-\gamma}.$$

We also have that:

$$L_\rho \geq L - \lceil 2\rho T_\alpha + \sqrt{8\rho T_\alpha \log(1/\delta)} \rceil \geq L - 2\rho T_\alpha - \sqrt{8\rho T_\alpha \log(1/\delta)} - 1$$

where the first inequality follows from the definition of $L_\rho$, the second from the fact that $\lceil n \rceil \leq n+1$. Therefore, defining $L'_\rho = L - 2\rho T_\alpha - \sqrt{8\rho T_\alpha \log(1/\delta)} - 1$, gives us that for any $0 < \rho < 1/2$:

$$\sum_{t=T_\alpha-L_\rho}^{T_\alpha} \gamma_t \geq (\gamma^{-L'_\rho}-1)\frac{\gamma^{T_\alpha}}{1-\gamma}.$$

Selecting $\rho = 1/(8T_\alpha \log(1/\delta))$ gives us:

$$\rho\left(\gamma^{-L'_\rho}-1\right)\frac{\gamma^{T_\alpha}}{1-\gamma} \geq (8\log(1/\delta))^{-1}\frac{1}{T_\alpha}\left(\gamma^{-L+3}-1\right)\frac{\gamma^{T_\alpha}}{1-\gamma},$$

which completes the lemma. $\qquad\square$

### A.4 Proof of Lemma 3

*Proof.* Let $S_1$ and $S_2$ be the excess surplus that a surplus-maximizing buyer earns over the truthful buyer during the learning and exploit phase of the LEAP algorithm, respectively. We have

$$S_2 \leq \sum_{t=T_\alpha+1}^{T} \gamma^{t-1} = \gamma^{T_\alpha}\sum_{t=0}^{T-T_\alpha-1}\gamma^t = \frac{\gamma^{T_\alpha}}{1-\gamma}(1-\gamma^{T-T_\alpha}). \tag{7}$$

Indeed, this an upper bound on the total surplus any buyer can hope to achieve in the second phase. Now observe that for any constants $C > 0$, $\delta_0 > 0$ and $\rho^*$ as defined in equation (6), we have

$$
\begin{aligned}
E[S_1] &= \Pr[\mathcal{E}_{\delta_0,\rho^*} \wedge L \geq C]E[S_1 \mid \mathcal{E}_{\delta_0,\rho^*} \wedge L \geq C] + \Pr[\neg\mathcal{E}_{\delta_0,\rho^*} \vee L < C]E[S_1 \mid \neg\mathcal{E}_{\delta_0,\rho^*} \vee L < C] \\
&\leq \Pr[\mathcal{E}_{\delta_0,\rho^*} \wedge L \geq C]E[S_1 \mid \mathcal{E}_{\delta_0,\rho^*} \wedge L \geq C] \\
&= \Pr[\mathcal{E}_{\delta_0,\rho^*}]\Pr[L \geq C \mid \mathcal{E}_{\delta_0,\rho^*}]E[S_1 \mid \mathcal{E}_{\delta_0,\rho^*} \wedge L \geq C] \\
&\leq -(1-\delta_0)\Pr[L \geq C \mid \mathcal{E}_{\delta_0,\rho^*}]\frac{\gamma^{-C+3}-1}{8T_\alpha \log(1/\delta_0)}\left(\frac{\gamma^{T_\alpha}}{1-\gamma}\right)
\end{aligned}
$$

The steps follow respectively by the law of iterated expectation; because $S_1 \leq 0$ with probability 1, since the truthful buyer strategy gives maximal revenue during the non-adaptive first phase; definition of conditional probability; and finally, applying Lemma 8 to lower bound $\Pr[\mathcal{E}_{\delta_0,\rho^*}]$ and the second half of the proof of Lemma 2 (shown in Section A.3) to upper bound $E[S_1 \mid \mathcal{E}_{\delta_0,\rho^*} \wedge L \geq C]$ (which is a negative quantity).

Note, since we are assuming a surplus maximizing buyer, it must be the case that $0 \leq E[S_1 + S_2]$. Thus, using the upper bound on $S_2$ and the upper bound on $E[S_1]$, we can rewrite the fact $0 \leq E[S_1 + S_2]$ as:

$$Pr[L \geq C \mid \mathcal{E}_{\delta_0,\rho^*}](1-\delta_0)\frac{\gamma^{-C+3}-1}{8T_\alpha \log(1/\delta_0)}\left(\frac{\gamma^{T_\alpha}}{1-\gamma}\right) \leq \frac{\gamma^{T_\alpha}}{1-\gamma}(1-\gamma^{T-T_\alpha})$$

$$\iff Pr[L \geq C \mid \mathcal{E}_{\delta_0,\rho^*}] \leq 8T_\alpha \log(1/\delta_0)(1-\gamma^{T-T_\alpha})/((1-\delta_0)(\gamma^{-C+3}-1))$$

Therefore, when

$$C = \frac{\log\left(\frac{(1-\gamma^{T-T_\alpha})8T_\alpha \log(1/\delta_0)}{\delta_0(1-\delta_0)}+1\right)}{\log(1/\gamma)} - 3 \qquad \text{we have} \qquad \Pr[L \geq C \mid \mathcal{E}_{\delta_0,\rho^*}] \leq \delta_0.$$

Fixing this choice of $C$, lets us conclude:

$$\Pr[L \geq C] = \Pr[L \geq C \mid \mathcal{E}_{\delta_0, \rho^*}] \Pr[\mathcal{E}_{\delta_0, \rho^*}] + \Pr[L \geq C \mid \neg \mathcal{E}_{\delta_0, \rho^*}] \Pr[\neg \mathcal{E}_{\delta_0, \rho^*}]$$
$$\leq \Pr[L \geq C \mid \mathcal{E}_{\delta_0, \rho^*}] + \Pr[\neg \mathcal{E}_{\delta_0, \rho^*}] \leq \delta_0 + \delta_0$$

Thus, setting $\delta_0 = \delta/2$ tells us that $\Pr[L < C] \geq 1 - \delta$. Finally, to complete the lemma, we upper bound $C$ by dropping the terms $(1 - \gamma^{T - T_\alpha})$ and $-3$, and using $1/(\delta_0(1 - \delta_0)) = 2/(\delta(1 - \delta/2)) \leq 4/\delta$. $\qquad\square$

## A.5 Results from Rakhlin et al. [14]

Let $Z_t = (\nabla F(\mathbf{w}_t) - \mathbf{g}_t)^\top (\mathbf{w}_t - \mathbf{w}^*)$ and

$$Z(T) = \frac{2}{\lambda} \sum_{t=2}^{T} \frac{Z_t}{t} \prod_{t'=t+1}^{T} \left(1 - \frac{2}{t'}\right). \tag{8}$$

Rakhlin et al. [14] proved the following upper bound on $Z(T)$ in the last half of the proof of their Proposition 1. For convenience, we isolate it into a separate lemma.

**Lemma 9.** *Let $\mathbf{w}_1, \ldots, \mathbf{w}_T$ be any sequence of weight vectors. If $E[\mathbf{g}_t] = \nabla F(\mathbf{w}_t)$ and $\|\mathbf{g}_t\|^2 \leq G^2$ then for any $\delta < 1/e$ and $T \geq 2$*

$$Z(T) \leq \frac{16 G \sqrt{\log(\log(T)/\delta)}}{\lambda(T-1)T} \sqrt{\sum_{t=2}^{T} (t-1)^2 \|\mathbf{w}_t - \mathbf{w}^*\|^2} + \frac{16 G^2 \log(\log(T)/\delta)}{\lambda^2 T}.$$

Importantly, for the previous lemma to hold it is *not* necessary for the $\mathbf{w}_t$'s to have been generated by stochastic gradient descent. The same remark applies to the next lemma, which gives a recursive upper bound on $\|\mathbf{w}_{t+1} - \mathbf{w}^*\|^2$, and which was also proven by Rakhlin et al. [14] in the last half of the proof of their Proposition 1.

**Lemma 10.** *Let $\mathbf{w}_1, \ldots, \mathbf{w}_{T+1}$ be any sequence of weight vectors. Suppose the following three conditions hold:*

1. *$\|\mathbf{w}_t - \mathbf{w}^*\|^2 \leq \frac{a}{t}$ for $t \in \{1, 2\}$,*

2. *$\|\mathbf{w}_{t+1} - \mathbf{w}^*\|^2 \leq \frac{b}{(t-1)t} \sqrt{\sum_{i=2}^{t} (i-1)^2 \|\mathbf{w}_i - \mathbf{w}^*\|^2} + \frac{c}{t}$ for $t \in \{2, \ldots, T\}$, and*

3. *$a \geq \frac{9b^2}{4} + 3c$.*

*Then $\|\mathbf{w}_{T+1} - \mathbf{w}^*\|^2 \leq \frac{a}{(T+1)}$.*

## A.6 Proof of Lemma 4

*Proof.* Recall that $F$ is $\lambda$-strongly convex. A well-known property of $\lambda$-strongly convex functions is that

$$\nabla F(\mathbf{w}')^\top (\mathbf{w}' - \mathbf{w}'') \geq F(\mathbf{w}') - F(\mathbf{w}'') + \frac{\lambda}{2} \|\mathbf{w}' - \mathbf{w}''\|^2 \tag{9}$$

for any weight vectors $\mathbf{w}', \mathbf{w}''$ (for example, see [15]). Letting $\mathbf{w}' = \mathbf{w}^*$ and $\mathbf{w}'' = \mathbf{w}$ in Eq. (9) we have

$$0 = \nabla F(\mathbf{w}^*)^\top (\mathbf{w}^* - \mathbf{w}) \geq F(\mathbf{w}^*) - F(\mathbf{w}) + \frac{\lambda}{2} \|\mathbf{w}^* - \mathbf{w}\|^2$$

$$\Rightarrow F(\mathbf{w}) - F(\mathbf{w}^*) \geq \frac{\lambda}{2} \|\mathbf{w}^* - \mathbf{w}\|^2 \tag{10}$$

where we used the fact that $\mathbf{w}^*$ minimizes $F$, and thus $\nabla F(\mathbf{w}^*) = \mathbf{0}$. Now letting $\mathbf{w}' = \mathbf{w}$ and $\mathbf{w}'' = \mathbf{w}^*$ in Eq. (9) and applying Eq. (10) proves

$$\nabla F(\mathbf{w})^\top (\mathbf{w} - \mathbf{w}^*) \geq \lambda \|\mathbf{w} - \mathbf{w}^*\|^2. \tag{11}$$

Note that $\tilde{\mathbf{g}}_t = \mathbf{g}_t \pm \mathbf{1}\{t \in \mathcal{L}\}\mathbf{x}_t$, where the $\pm$ depends on the value of $a_t$. Let $Z_t = (\nabla F(\mathbf{w}_t) - \mathbf{g}_t)^\top (\mathbf{w}_t - \mathbf{w}^*)$. We have

$$
\begin{aligned}
\|\mathbf{w}_{t+1} - \mathbf{w}^*\|^2 &= \|\mathbf{w}_t - \eta_t \tilde{\mathbf{g}}_t - \mathbf{w}^*\|^2 \\
&= \|\mathbf{w}_t - \mathbf{w}^*\|^2 - 2\eta_t \tilde{\mathbf{g}}_t^\top (\mathbf{w}_t - \mathbf{w}^*) + \eta_t^2 \|\tilde{\mathbf{g}}_t\|^2 \\
&= \|\mathbf{w}_t - \mathbf{w}^*\|^2 - 2\eta_t \mathbf{g}_t^\top (\mathbf{w}_t - \mathbf{w}^*) \pm 2\eta_t \mathbf{1}\{t \in \mathcal{L}\}\mathbf{x}_t^\top (\mathbf{w}_t - \mathbf{w}^*) + \eta_t^2 \|\tilde{\mathbf{g}}_t\|^2 \\
&\leq \|\mathbf{w}_t - \mathbf{w}^*\|^2 - 2\eta_t \mathbf{g}_t^\top (\mathbf{w}_t - \mathbf{w}^*) + 4\eta_t \mathbf{1}\{t \in \mathcal{L}\} + \eta_t^2 G^2 \quad\quad (12) \\
&= \|\mathbf{w}_t - \mathbf{w}^*\|^2 - 2\eta_t \nabla F(\mathbf{w}_t)^\top (\mathbf{w}_t - \mathbf{w}^*) + 2\eta_t Z_t + 4\eta_t \mathbf{1}\{t \in \mathcal{L}\} + \eta_t^2 G^2 \\
&\leq \|\mathbf{w}_t - \mathbf{w}^*\|^2 - 2\eta_t \lambda \|\mathbf{w}_t - \mathbf{w}^*\|^2 + 2\eta_t Z_t + 4\eta_t \mathbf{1}\{t \in \mathcal{L}\} + \eta_t^2 G^2 \quad\quad (13) \\
&= (1 - 2\lambda\eta_t) \|\mathbf{w}_t - \mathbf{w}^*\|^2 + 2\eta_t Z_t + 4\eta_t \mathbf{1}\{t \in \mathcal{L}\} + \eta_t^2 G^2
\end{aligned}
$$

where in Eq. (12) we used $\mathbf{x}_t^\top (\mathbf{w}_t - \mathbf{w}^*) \leq \|\mathbf{x}_t\| \|\mathbf{w}_t - \mathbf{w}^*\| \leq 2$ and $\|\tilde{\mathbf{g}}_t\|^2 \leq G^2$. In Eq. (13) we used Eq. (11). For any $T' \in \{2, \ldots, T_\alpha\}$ let $Y_t(T') = \prod_{t'=t+1}^{T'} (1 - 2\lambda\eta_{t'})$. Unrolling the above recurrence till $t = 2$ yields

$$
\|\mathbf{w}_{T'+1} - \mathbf{w}^*\|^2 \leq Y_1(T') \|\mathbf{w}_2 - \mathbf{w}^*\|^2 + 2\sum_{t=2}^{T'} \eta_t Z_t Y_t(T') + 4\sum_{t=2}^{T'} \eta_t \mathbf{1}\{t \in \mathcal{L}\}Y_t(T') + G^2 \sum_{t=2}^{T'} \eta_t^2 Y_t(T').
$$

Now substitute $\eta_t = \frac{1}{\lambda t}$, and note that since $(1 - 2\lambda\eta_2) = 0$ and $T' \geq 2$ we have $Y_1(T') = 0$, so the first term is zero. Also the second term is equal to $Z(T')$ by the definition in Eq. (8) in Appendix A.5. Simplifying leads to

$$
\|\mathbf{w}_{T'+1} - \mathbf{w}^*\|^2 \leq Z(T') + \frac{4}{\lambda}\sum_{t=2}^{T'} \mathbf{1}\{t \in \mathcal{L}\}\frac{Y_t(T')}{t} + \frac{G^2}{\lambda^2}\sum_{t=2}^{T'} \frac{Y_t(T')}{t^2}. \quad\quad (14)
$$

Now observe that for $t \geq 2$

$$
\log Y_t(T') = \sum_{t'=t+1}^{T'} \log\left(1 - \frac{2}{t'}\right) \leq -2\sum_{t'=t+1}^{T'} \frac{1}{t'} = -2\left(\sum_{t'=1}^{T'} \frac{1}{t'} - \sum_{t'=1}^{t} \frac{1}{t'}\right) \leq -2(\log T' - \log t - 1),
$$

where the last inequality uses a lower bound on the $t$-th harmonic number and upper bound on the $T'$-th harmonic number. Thus, $Y_t(T') \leq \frac{e^2 t^2}{T'^2}$ and plugging back into Eq. (14) yields

$$
\|\mathbf{w}_{T'+1} - \mathbf{w}^*\|^2 \leq Z(T') + \frac{4e^2}{\lambda T'^2}\sum_{t=2}^{T'} \mathbf{1}\{t \in \mathcal{L}\}t + \frac{e^2 G^2}{\lambda^2 T'} \leq Z(T') + \frac{4e^2 L}{\lambda T'} + \frac{e^2 G^2}{\lambda^2 T'}.
$$

where the second inequality follows from $\sum_{t=2}^{T'} \mathbf{1}\{t \in \mathcal{L}\}t \leq LT'$. Now, to bound the term $Z(T')$, we apply Lemma 9 from Appendix A.5 and conclude that for $\delta \in [0, 1/e]$, with probability at least $1 - \delta$, for all $T' \in \{2, \ldots, T_\alpha\}$

$$
Z(T') \leq \frac{16G\sqrt{\log(\log(T')/\delta)}}{\lambda(T'-1)T'}\sqrt{\sum_{t=2}^{T'} (t-1)^2 \|\mathbf{w}_t - \mathbf{w}^*\|^2} + \frac{16G^2 \log(\log(T')/\delta)}{\lambda^2 T'}.
$$

Plugging this back in and simplifying we get, with probability at least $1 - \delta$, for all $T' \in \{2, \ldots, T_\alpha\}$

$$
\|\mathbf{w}_{T'+1} - \mathbf{w}^*\|^2 \leq
$$

$$
\frac{16G\sqrt{\log(\log(T')/\delta)}}{\lambda(T'-1)T'}\sqrt{\sum_{t=2}^{T'} (t-1)^2 \|\mathbf{w}_t - \mathbf{w}^*\|^2} + \frac{1}{T'}\left(\frac{(16\log(\log(T')/\delta) + e^2)G^2}{\lambda^2} + \frac{4e^2 L}{\lambda}\right).
$$

In order to apply Lemma 10 in Appendix A.5 let

$$
a = \frac{(624\log(\log(T_\alpha)/\delta) + e^2)G^2}{\lambda^2} + \frac{4e^2 L}{\lambda},
$$

$$
b = \frac{16G\sqrt{\log(\log(T')/\delta)}}{\lambda}, \text{ and}
$$

$$
c = \frac{(16\log(\log(T')/\delta) + e^2)G^2}{\lambda^2} + \frac{4e^2 L}{\lambda}.
$$

It is a straightforward calculation to show that $a \geq \frac{9b^2}{4} + 3c$. Also for any $T'$

$$G\|\mathbf{w}_{T'} - \mathbf{w}^*\| \geq \|\nabla F(\mathbf{w}_{T'})\| \|\mathbf{w}_{T'} - \mathbf{w}^*\| \geq \nabla F(\mathbf{w}_{T'})^\top (\mathbf{w}_{T'} - \mathbf{w}^*) \geq \lambda \|\mathbf{w}_{T'} - \mathbf{w}^*\|^2$$

where the last inequality follows from Eq. (11). Dividing both sides by $\lambda \|\mathbf{w}_{T'} - \mathbf{w}^*\|$ proves $\|\mathbf{w}_{T'} - \mathbf{w}^*\| \leq \frac{G}{\lambda}$ for all $T'$, which implies $\|\mathbf{w}_{T'} - \mathbf{w}^*\|^2 \leq a/T'$ for $T' \in \{1, 2\}$. Now we can apply Lemma 10 in Appendix A.5 to show

$$\|\mathbf{w}_{T_\alpha+1} - \mathbf{w}^*\|^2 \leq \frac{1}{T_\alpha + 1}\Big(\frac{(624\log(\log(T_\alpha)/\delta) + e^2)G^2}{\lambda^2} + \frac{4e^2 L}{\lambda}\Big),$$

which completes the proof. $\qquad\square$

## A.7 Proof of Proposition 2

*Proof.* We will use an inductive argument. Note that, before the projection step $\beta_1 = 2a_1/\lambda$ and after projection $\beta_1 = a_1/\sqrt{K(\mathbf{x}_1, \mathbf{x}_1)}$. Thus, $\mathbf{w}_1 = \mathbf{0}$ and $\mathbf{w}_2 = \beta_1\phi(\mathbf{x}_1) = \frac{a_1}{\sqrt{K(\mathbf{x}_1, \mathbf{x}_1)}}\phi(\mathbf{x}_1)$ match the hypotheses returned by the LEAP algorithm when operating in the feature space induced by $\phi(\cdot)$ and using the projection $\Pi_{\mathcal{W}}$ for $\mathcal{W} = \{\mathbf{w} : \|\mathbf{w}\|_2 \leq 1\}$. Now, assuming the inductive hypothesis, we have $\mathbf{w}_t = \sum_{i=1}^{t-1} \beta_i \phi(\mathbf{x}_i)$ and we have, before projection,

$$\sum_{i=1}^{t} \beta_i \phi(\mathbf{x}_i) = \mathbf{w}_t + \beta_t = \mathbf{w}_t - \frac{2}{\lambda t}\Big(\sum_{i=1}^{t-1} \beta_i K(\mathbf{x}_i, \mathbf{x}_t) - a_t\Big)\phi(\mathbf{x}_t) = \mathbf{w}_t - \frac{2}{\lambda t}(\mathbf{w}_t^\top \phi(\mathbf{x}_t) - a_t)\phi(\mathbf{x}_t)$$

and, after projection,

$$\frac{\sum_{i=1}^{t} \beta_i \phi(\mathbf{x}_i)}{\sqrt{\sum_{i,j=1}^{t} \beta_i \beta_j K(\mathbf{x}_i, \mathbf{x}_j)}} = \frac{\sum_{i=1}^{t} \beta_i \phi(\mathbf{x}_i)}{\|\sum_{i=1}^{t} \beta_i \phi(\mathbf{x}_i)\|} = \frac{\mathbf{w}_t - \frac{2}{\lambda t}(\mathbf{w}_t^\top \phi(\mathbf{x}_t) - a_t)\phi(\mathbf{x}_t)}{\|\mathbf{w}_t - \frac{2}{\lambda t}(\mathbf{w}_t^\top \phi(\mathbf{x}_t) - a_t)\phi(\mathbf{x}_t)\|}$$

$$= \Pi_{\mathcal{W}}\Big(\mathbf{w}_t - \frac{2}{\lambda t}(\mathbf{w}_t^\top \phi(\mathbf{x}_t) - a_t)\phi(\mathbf{x}_t)\Big) = \mathbf{w}_{t+1}$$

which proves the equivalence of the first phase of the two algorithms in the feature space induced by $\phi(\cdot)$. Note, in the second phase neither $\boldsymbol{\beta}$ or $\mathbf{w}_{T_\alpha+1}$ is updated, and from the preceding argument we have

$$p_t = \sum_{i=1}^{T_\alpha} \beta_i K(\mathbf{x}_i, \mathbf{x}_t) - \epsilon = \Big(\sum_{i=1}^{T_\alpha} \beta_i \phi(\mathbf{x}_i)\Big)\phi(\mathbf{x}_t) - \epsilon = \mathbf{w}_{T_\alpha+1}^\top \phi(\mathbf{x}_t) - \epsilon,$$

which shows the equivalence of the two algorithms in the second phase as well.

The bound $\|\mathbf{w}_t\| \leq 1$ follows directly from the definition of the projection $\Pi_K$. Using $\mathbf{w}_t = \sum_{i=1}^{t-1} \beta_i \phi(\mathbf{x}_i)$, we have that the gradient is

$$\mathbf{g}_t = 2(\mathbf{w}_t^\top \phi(\mathbf{x}_t) - a_t)\phi(\mathbf{x}_t) = 2\Big(\sum_{i=1}^{t} \beta_i \phi(\mathbf{x}_i)^\top \phi(\mathbf{x}_t) - a_t\Big)\phi(\mathbf{x}_t).$$

Finally, we can bound $\|\mathbf{g}_t\| \leq 2(|\mathbf{w}_t^\top \phi(\mathbf{x}_t)| + 1)\|\phi(\mathbf{x}_t)\| \leq 2(\|\mathbf{w}_t\|\|\phi(\mathbf{x}_t)\| + 1) \leq 4$, which follows from $\|\mathbf{w}_t\| \leq 1$ and $\|\phi(\mathbf{x}_t)\| = \sqrt{K(\mathbf{x}_t, \mathbf{x}_t)} \leq 1$. $\qquad\square$

## A.8 Doubling trick

**Corollary 2.** *Partition all $T$ rounds into $\lceil \log_2 T \rceil$ consecutive phases, where each phase $i$ has length $T_i = 2^i$. Run an independent instance of the LEAP algorithm in each phase, tuning $\epsilon$ and $\alpha$ as in Theorem 2, using horizon length $T_i$. Against a surplus-maximizing buyer, the seller's regret against a surplus-maximizing buyer is $R(T) \leq O\big(T^{2/3}\sqrt{\frac{\log(T)}{\log(1/\gamma)}}\big)$.*

*Proof.* Since an independent instance of the algorithm is run in each phase, the buyer will behave so as to maximize surplus in each phase independently, without regard to what occurs in other phases.

Moreover, the discount factor for the $s$th round in any phase $i$ is $\gamma^{t_i+s} = \gamma^{t_i}\gamma^s$, where $t_i$ is the first round of phase $i$. It is easy to see that the behavior of a surplus-maximizing buyer is unchanged if we scale her surplus in every round by a constant. Therefore the analysis of Theorem 2 is directly applicable to every phase, and we can combine the analysis for all phases using the doubling trick, as follows.

Let $R_i$ be the seller's strategic regret in phase $i$ and $n = \lceil \log_2 T \rceil$. By Theorem 2 there exists a constant $C$ depending only on $\lambda$ such that

$$R(T) = \sum_{i=1}^{\lceil \log_2 T \rceil} R_i \leq \frac{C}{\sqrt{\log(1/\gamma)}} \sum_{i=1}^{\lceil \log_2 T \rceil} T_i^{2/3}\sqrt{\log_2 T_i} = \frac{C}{\sqrt{\log(1/\gamma)}} \sum_{i=1}^{\lceil \log_2 T \rceil} \left(2^{2/3}\right)^i \sqrt{i} \quad (15)$$

Let $S_{r,n} = \sum_{i=1}^{n} r^i \sqrt{i}$. Observe that

$$S_{r,n+1} = \sum_{i=1}^{n+1} r^i \sqrt{i} = r^{n+1}\sqrt{n+1} + \sum_{i=1}^{n} r^i \sqrt{i} = r^{n+1}\sqrt{n+1} + S_{r,n}$$

and

$$S_{r,n+1} = r\sum_{i=1}^{n+1} r^{i-1}\sqrt{i} \geq r\sum_{i=1}^{n+1} r^{i-1}\sqrt{i-1} = r\sum_{i=2}^{n+1} r^{i-1}\sqrt{i-1} = r\sum_{i=1}^{n} r^i\sqrt{i} = rS_{r,n}$$

Combining the previous two inequalities proves $r^{n+1}\sqrt{n+1} + S_{r,n} \geq rS_{r,n}$, which can be rearranged to show

$$\sum_{i=1}^{n} r^i\sqrt{i} \leq \frac{r^{n+1}\sqrt{n+1}}{r-1}.$$

Applying the above inequality for $n = \lceil \log_2 T \rceil$ and $r = 2^{2/3}$ proves

$$\sum_{i=1}^{\lceil \log_2 T \rceil} \left(2^{2/3}\right)^i \sqrt{i} \leq \frac{(2^{2/3})^{\lceil \log_2 T \rceil + 1}\sqrt{\lceil \log_2 T \rceil + 1}}{2^{2/3}-1}$$

$$\leq \frac{(2^{2/3})^{\log_2 T + 2}\sqrt{\log_2 T + 2}}{2^{2/3}-1}$$

$$= \frac{2^{4/3}}{2^{2/3}-1}T^{2/3}\sqrt{\log_2 T + 2}.$$

Combining the above with Eq (15) proves the corollary. $\qquad\square$