[Reviews · NeurIPS 2014]

Submitted by Assigned_Reviewer_26

This paper introduces the problem of repeated auctions with contextual information. Specifically, the auctioneer interacts with a single agent (at least in their initial model) who has a context vector x, corresponding to observable attributes, and a privately held weight vector w, such that the agent’s value for the good is x*w. The goal of the auctioneer is to maximize the revenue from the posted price auction by learning w over time. However, the agent understand that the auctioneer may be adjusting prices based off of the his purchasing behavior, so he may lie (that is, not purchase the product even though it would give him positive surplus) in order to lower the price in future rounds. The authors present an algorithm based on stochastic gradient descent. The authors prove that the number of “lies” by an agent is mitigated b the agent’s discount factor for future utility and use existing results to show that these bounded number of lies do not affect the learning algorithm too much, thereby bounding the “strategic regret” in this model.

I like this paper and I view its main contribution as introducing the use of contextual information to repeated auctions. The authors make a good point that the reason that so many auctions on the web have only a single participant is because of targeting, so it seems natural that the mechanism itself should be able to use targeted information (i.e. context) to set prices.I think this model (or at least this family of models) is very compelling and raises a lot of interesting questions. This paper does a good job at beginning to address this model and will inspire more work to be done on this area.

The technical strength of this paper is solid, primarily consisting of extending existing techniques to the details of this model (although this is not a trivial task) and the exposition is relatively clear. The technical strength could have been improved via discussion of regret lower bounds in this model. It would have been nice to have a deeper understanding of regret lower bounds in this model versus similar models without context (e.g. Amin et al 2013). However this seems like significant extra work and is reasonably outside the scope of this paper.

I found the section on multiple buyers a little unconvincing, which is not to imply that I believe it is wrong, but rather that the extension from posted-pricing to a reserve price auction is not straightforward and warrants more exposition/exploration. Its not obvious how the LEAP algorithm extends in the case of multiple buyers during the exploit phase. Presumably, the algorithm realizes the context vectors for all agents and then posts a reserve price of max_{i in N} x_i * w_t but this is not actually stated. As the authors note, the “market value” assumption seems a little odd; its not clear that one can simply convert between the two settings (that is, taking the maximum of a number of agents versus just acting as if there were one agent that always had the maximum value). Personally, I’m more interested in expanding this case than I am in the extension of non-linear valuation functions and I wish the paper had expanded more on these details rather than showing kernelized extension.

Beyond these small confusions, it seems that the multiple bidder setting has more complexity and nuance than presented in this paper. For example, it seems that more buyers should decrease regret in a number of ways. First, more buyers should increase the effect of the discount factor because, all else being equal, a buyer will expect to wait a longer time to purchase in the future (because now he needs to take into consideration the probability that he has the highest valuation amongst the group of agents that show up in any given round). Under the assumption of IID valuations, this should increase his expected wait time by a factor K (the number of players), hence decreasing his future utility exponentially (due to discounting). So it seems that lying should become less of an issue.

In terms of related work, the authors could connect better to the study of contextual multi-armed bandits as the settings are similar.

In summary, I think this work is a valuable contribution because this is a good model, in the sense that is practically relevant and is an interesting example of the intersection of machine learning and algorithmic game theory. The case of multiple bidders case is more complex than the current work considers and deserves future work but the work is technically strong enough with the single agent model.
Summary: A solid paper that introduces a very compelling model at the intersection of machine learning and algorithmic game theory. While some of the technical work is not the strongest, this is a paper that is likely to inspire future work.

Submitted by Assigned_Reviewer_30

Update after author response:

It seems like the authors will be able to fix the mistake in Lemmas 2,3, so I am changing my recommendation accordingly.

Repeated Contextual Auctions with Strategic Buyers

The paper describes a scheme for setting prices in a repeated contextual
auction in which the buyer's value depends linearly on the context, and
shows that this scheme achieves O(T^{2/3}) regret (up to log-factors)
when the buyer(s) either are truthful about their value or maximize
their discounted surplus, with later rounds discounted exponentially.
Given the financial stakes for certain companies, there should be
significant interest in this problem.

I take it from the introduction that this is pioneering work for the
contextual setting with a surplus-maximizing buyer. It is therefore
perhaps not surprising that some of the modelling assumptions do not
appear to be very realistic. (One needs to start somewhere when
analysing a new setting.)

The paper is mostly well written, but there is one significant mistake
in the derivations leading up to the main theorem (Thm 2), as well as
some minor errors that should be easy to fix. I hope that in their
rebuttal the authors can convincingly explain how they would fix the
significant mistake.

Significant mistake:

Lemmas 2 and 4 assume that the number of lies L is independent of
the sequence of prices p_t and values v_t, but, in Lemma 3 and Thm 2, L is a
random variable that can depend on p_t and v_t, so the results do not
line up. In particular, line 592 is not correct, because conditioning on
L might change the distribution of p_t and v_t and is therefore not the
same as fixing L independently in advance. There must be a similar issue
in the proof of Thm 2 somewhere.

Unrealistic modelling assumptions:

Since reviews will become public, I think there is some value in listing
some critical remarks about the paper's modelling assumptions, even
though I would not expect the authors to change anything based on these
comments.

* Exponential discount factors mean that one unit of revenue today is
about equally valuable as one unit of revenue on all days in the
future together. This seems rather steep. And there might be room for
improvement, because the lower bound in line 229 would not be too bad
if, e.g., gamma_t = 1/t.

* In Section 5, p.8, the authors introduce a linear model for the market
valuation of a good, which is the maximum of the buyers' valuations.
Although they defend this choice in lines 373-377 by presenting it
from a different perspective, I still find it rather implausible that
a maximum would be a linear function. In line with their previous
modelling of each buyer's valuation as a linear function, we should
rather expect the maximum to be a convex function.

Minor mistakes:

* line 157: v_t = w*^T x_t is not the same as w_t^T x_t, but w_t^T x_t
<= 1 still holds by Cauchy-Schwarz.
* line 507: skip the first equality, which appears to be incorrect and
go directly to the second
* line 527: M should be defined with the opposite inequality: "... <=
rho}". In the rest of the proof, be consistent about whether you take
this inequality to be strict or not.
* line 537: '8' should be '4' based on Lemma 7.
* line 544: there is no need to round up M, as it is already an integer.
* line 549: a factor rho is missing in the last term.
* line 559: there is no need to round up the probabilistic upper bound
on M, because there is no reason it needs to be an integer.

Other comments:

* line 98: w*^T x <= 1 already follows from Cauchy-Schwarz, so it is not
an extra assumption.
* line 117: O(T^{2/3}sqrt{log(T*log(T))}) can be simplified to O(T^{2/3}sqrt{log(T*)})
* The LEAP algorithm: lambda is also a parameter that needs to be set
correctly, as discussed on p.4.
* Thm 1: Explicitly add the assumption that the minimum eigenvalue of
2E[xx^T] is at least lambda > 0. It is easy to miss when it is only
stated in the text.
* Line 592: please explicitly add the event with probability at most
delta that you are dropping. You need to motivate that the
corresponding conditional expectation is negative.

Summary: Given the financial stakes for certain companies, there should be significant interest in this problem, although the modelling assumptions do not appear to be directly realistic. The paper is mostly well written, but there is one significant mistake. UPDATE: It seems that the authors will be able to fix the mistake, so I have adjusted my scores accordingly.

Submitted by Assigned_Reviewer_38

The paper studies revenue-maximization in a repeated posted-price setting where at each round the seller announces a price and the buyer either accepts or rejects the offer. The problem is motivated by ad exchange in the web (second-price auctions). Two types of buyers are considered: truthful (accepts if the offer is higher than his value) and surplus-maximizing (that might reject good offers to get more profitable offers in the future). The goal of the learner is to have small regret, defined by the difference between the total revenue and the revenue that could be obtained had the seller chosen to offer the optimal price (buyer’s value).

This problem was studied earlier by [1]. The contribution of the present paper is to extend the setting of [1] to the case when the value is an unknown linear function of a context vector. The authors propose a forced exploration algorithm that has a learning phase followed by an exploitation phase. In the learning phase, the seller offers uniformly randomly selected prices, and updates a parameter vector using a gradient rule.

When the buyer is truthful, it is shown that the proposed gradient is an unbiased estimate of the gradient of a surrogate loss. This is a nice observation in the paper. Then, it is shown that the algorithm enjoys a O(T^{2/3}) regret bound against truthful buyers. But it is not clear if this rate is optimal.

A similar rate is shown against the surplus-maximizing buyers, but only when the buyer maximizes his cumulative discounted surplus. [1] shows that the problem is hopeless when discount factor is one. Still, this is a strange 2 player game as one objective is discounted and the other is not.

The main contribution of the paper is extending results of [1] to the contextual setting. The paper can be improved by adding some interesting experiments. Overall, the problem is interesting and the proposed algorithm is simple, but no numerical experiments are reported. The paper is well-written.

Typos and minor issues:

* Page 2, line 97: w_t^* → w^*
* Page 3, line 150: w^* → w_*
* Page 3, line 152, … where G^2 = 4: Why defining a new variable G?!
* Page 3, line 154: x_t is missing in the gradient estimates
* Page 4, line 157: w_t → w_*
* Page 6, line 300: “against” should be removed
Summary: The problem is interesting and the proposed algorithm is simple, but no numerical experiments are reported. The paper is well-written.
Author Feedback
Author rebuttal: The most serious issue is raised by Reviewer_30, who found a mistake in the proof of Lemma 3. We agree that the current argument is incorrect, but believe that the correction is minor. In the following, we give an informal description of how the proof of Lemma 3 is corrected, followed by a more detailed treatment.

We first restate Lemma 2, which is used to prove Lemma 3. While Lemma 2 is correct, we use a more refined version of it to address the reviewer's concern. Informally, there is an event, call it "Z", which occurs with probability at least 1-delta, namely that |v_t - p_t| is large on the vast majority of rounds. Lemma 2 shows two things: (1) Z occurs with probability at least 1-delta, and (2) for any outcome in Z, it must be the case that a buyer who tells L lies loses the surplus claimed in Lemma 2 line 248.

The reviewer correctly observes that there is a problem in the proof of Lemma 3. The optimal buyer strategy is allowed to adapt to the sequence {v_t,p_t}. Therefore the number of lies told L is not independent of {v_t,p_t} or, for that matter, Z. However, we claim that the argument in Lemma 3 (which we address more formally below), does correctly establish Pr(L >= C | Z) <= delta (rather than Pr(L >= C) <= delta as previously claimed). The final step to completing the argument for Lemma 3 is observing that Pr(L >= C) = Pr(L >= C | Z)Pr(Z) + Pr(L >= C | ~Z)Pr(~Z) <= Pr(L >= C | Z) + Pr(~Z) <= 2\delta. This ultimately results in a factor of 64 inside the log term in the statement of Lemma 3, rather than 32.

More detailed correction:

Restatement of Lemma 2: Define the random variable M_rho = sum_{t = 1}^{T_alpha} 1{|v_t - p_t| < rho}, the number of times that the gap between the price offered, and the buyer's value, is smaller than rho during the learning phase. (There is a typo in the definition of M in the submitted paper). For delta > 0, let Z_{delta, rho} = {M_rho <= 2 rho T_alpha + 4 sqrt{rho T_alpha log(1/delta)}}, the event that there are not too many rounds on which this gap is smaller than rho. Let Z_delta = Z_{delta,rho*}, where rho* is set as in line 567. More carefully stated, Lemma 2 establishes (1) Pr(Z_delta) >= (1-delta), (2) Let L be a random variable that depends on {p_t,v_t}, which indicates the number of lies told by an arbitrary buyer strategy. On the event Z_delta, the surplus lost by such a buyer is bounded by the quantity on line 248.

Fixed Proof of Lemma 3: Let S_1 be the difference between the truthful buyer's and the surplus-maximizing buyer's surplus during the learning phase of the LEAP algorithm. Since LEAP is non-adaptive during the learning phase, on any sequence {p_t,v_t} the truthful strategy gains more revenue during the learning phase, and S_1 <= 0 with probability 1.

Fix delta0 > 0, and let C be a number to be chosen later.

E[S_1] = Pr(Z_delta0 and L >= C) E[S_1 | Z_delta0 and L >= C] +
Pr(~Z_delta0 or L < C) E[S_1 | ~Z_delta0 or L < c]
<= Pr(Z_delta0, L >= C) E[S_1 | Z_delta0, L >= C]
= Pr(Z_delta0) Pr(L >= C | Z_delta0) E[S_1 | Z_delta0, L >= C]
<= -(1-delta0) Pr(L >= C | Z_delta0) [Expression from Line 248]

Where these steps follow from the law of iterated expectation, because S_1 <= 0 always, properties of conditional expectation, and Lemma 2 resp.

Using the fact E[S_1 + S_2] <= 0 (since the buyer is surplus maximizing), and rearranging the equations lets us derive a C for which Pr(L >= C | Z_delta0) <= delta_0 (the value of C is the same as the one in line 600, but with the "delta (1-1/e)" term replaced with "delta0 (1 - delta0)"). Finally, Pr(L >= C) = Pr(L >= C | Z_delta0) Pr(Z_delta0) + Pr(L >= C | ~Z_delta0) Pr(~Z_delta0) <= Pr(L >= C | Z_delta0) + Pr(Z_delta0) <= 2 delta0, where the final inequality follows from our choice of C and Lemma 2 respectively. Setting delta0 = delta/2, and noting 1/(delta0(1-delta0)) = 2/(delta(1-delta/2)) <= 4/delta, changes the 32 in the log term in the statement of Lemma 3 to a 64.

With Lemma 3 fixed, we believe that the application of Lemma 4 in Theorem 2 is correct. We use the remaining space to address as many of the other concerns as possible.

Reviewer_{26,30} asks: How can the single-buyer setting can be a special case of the multiple-buyers setting?
We did not intend to claim that one is a special case of the other, although we understand that our presentation may have made this ambiguous.

Reviewer_30 on exponential discounting: "one unit of revenue today is about equally valuable as one unit of revenue on all days in the future together."
Actually if the discount rate is gamma the equivalence is: 1/(1-gamma) units of revenue today = one unit of revenue on all remaining days. Note that exponential discounting is a nearly ubiquitous assumption in the economics literature.

Reviewer_38: Why aren't there numerical experiments?
Evaluating an interactive, online price-setting algorithm is quite challenging, since it requires either testing the algorithm on live traffic from an e-commerce website or building a realistic simulator of buyer behavior, a difficult problem in itself. This is not to say that experiments are impossible or unnecessary -- in fact, we hope to do experiments on real data in the future -- just that we hope the reviewers will consider judging our paper on the merits of its theoretical contributions.

Reviewer_38: Why is the buyer's value discounted but the seller's is not?
In many important real-world markets, sellers are far more willing to wait for revenue than buyers are willing to wait for goods. For example, online advertisers are often interested in showing ads to users who have recently viewed their products online -- this practice is called 'retargeting' -- and the value of these user impressions decays rapidly over time. In this example the seller's discount rate is effectively 1 compared to the buyer's. Note this assumption has been made previously in the non-contextual setting.